

# Profiles of anemia in adolescent students with sports club membership in an outpatient clinic setting: a retrospective study

Kana Yamamoto[1,2], Morihito Takita[3], Masahiro Kami[4], Masaharu Tsubokura[5], Tetsuya Tanimoto[6], Toshio Kitamura[7] and Yoshinobu Takemoto[2]

[1] Department of Internal Medicine, The University of Tokyo, Bunkyo, Tokyo, Japan
[2] Department of Internal Medicine, Yoshinobu Clinic, Kagoshima, Japan
[3] Department of Internal Medicine, Navitas Clinic Tachikawa, Tachikawa, Tokyo, Japan
[4] Medical Governance Research Institute, Minato, Tokyo, Japan
[5] Department of Radiation Health Management, Fukushima Medical University, Fukushima, Japan
[6] Department of Internal Medicine, Navitas Clinic Kawasaki, Kawasaki, Kanagawa, Japan
[7] Division of Cellular Therapy, The Advanced Clinical Research Center, The Institute of Medical Science, The University of Tokyo, Minato, Tokyo, Japan

Corresponding author
Kana Yamamoto,
kanachan.y.0508@gmail.com

## ABSTRACT

**Background:** Anemia is a common health issue among adolescents. Anemic conditions could affect physical performance; however, the actual profiles of anemia in adolescent students in sports clubs have not been well documented.

**Methods:** We conducted a retrospective chart review of individuals aged 13–22 years who belonged to sports clubs in schools and visited an outpatient clinic between August 1, 2016, and August 31, 2020. The medical and laboratory records, including serum levels of ferritin, folate, vitamin B12, and creatinine kinase at their first visit were assessed.

**Results:** A total of 485 individuals (231 male (48%) and 254 female (52%) patients) were eligible for the study. The most common club activity was track and field ($n = 171$ (35%)). The overall prevalence of the World Health Organization-defined anemia was 16.5% (95% CI [13.1–20.4]; 9.0% [5.4–13.8] and 23.1% [17.8–29.2] in males and females, respectively) after excluding pre-treated individuals. Hypoferritinemia and elevation of serum creatinine kinase levels were identified as independent contributors to anemia in both sexes (odds ratios: 13.2 (95% CI [4.2–41.1]), $p < 0.001$ and 14.7 (95% CI [1.8–118.4]), $p = 0.012$, respectively for males; odds ratios: 6.6 (95% CI [1.3–13.9]), $p < 0.001$ and 2.7 (95% CI [1.4–5.5]), $p = 0.004$, respectively for females).

**Discussion:** Anemia is prevalent in both male and female adolescent students in sports clubs. Iron deficiency and excessive training indicated by elevated creatinine kinase levels may contribute to the risk of anemia. Physicians should assess the amount of exercise, and not merely iron storage, in clinical practice.

## INTRODUCTION

Anemia is a medical condition commonly observed in clinical practice, in which the number of red blood cells and, consequently, their oxygen-carrying capacity are insufficient to meet the physiological demands (*WHO, 2011*). It causes weakness, fatigue, difficulty concentrating, poor school performance, and decreased work productivity due to non-specific symptoms ascribed to the diminished levels of oxygen delivered to the body tissues. Anemia can be classified into three categories: defects in marrow production, red cell immaturity, and decreased red cell survival.

The most common cause of anemia is iron deficiency (ID), which causes a reduction in red cell production in the marrow (*Pasricha et al., 2021*). The clinical features of ID anemia (IDA) vary with sex and age. In adolescents, anemia is more common in females than in males. A meta-analysis of national surveys estimated the prevalence of IDA in women of pre-school and reproductive ages as 9.6% and 10.8%, respectively (*Petry et al., 2016*). Diet and lifestyle effects, owing to the desire for thinness in young women, and menstrual bleeding have been identified as the causes of IDA (*Fayet et al., 2014*). In one study, no significant sex difference was observed among older adults, and the prevalence of IDA was 12% (*Price et al., 2011*). The main causes of IDA include gastric cancer, peptic ulcers, gastrectomy, and aspirin medication (*Vetrano et al., 2020*).

Anemia is a serious problem among athletes (*McCormick et al., 2020*). The prevalence rates of IDA in *professional* and *elite* athletes were reported to be 2–29% and 3–7% in women (*Landahl et al., 2005*; *Ponorac et al., 2020*) and men (*Clement et al., 1987*; *Dubnov & Constantini, 2004*), respectively. The primary mechanisms by which sports lead to ID are increased iron loss and blockage of iron absorption due to hepcidin bursts (*Clenin et al., 2015*). Anemic conditions may affect the physical performance, and ID in anemia limits the transport of oxygen. Notably, ID without anemia also influences the oxygenation capacity. Hence, the athletes should be given regular opportunities to undergo clinical laboratory assessments, correction of nutritional iron intake, and nutritional fortification (*Sacirovic et al., 2013*). The International Olympic Committee 2009 Consensus Statement on periodic health evaluation of elite athletes recommends routine screening for ID (*Ljungqvist et al., 2009*).

ID occasionally causes problems other than anemia since iron is a major component of myoglobin, which is a protein found in the muscles (*Sim et al., 2019*). Athletes could easily become iron deficient due to their high volume of muscle mass if they are not supplemented by adequate iron. Some studies have demonstrated the beneficial effects of iron administration for the treatment of chronic heart failure and ID, even in the absence of anemia (*Anker et al., 2009*; *Okonko et al., 2008*). The prevalence rates of ID were reported to be 49–52% in female professional athletes (*Nabhan et al., 2020*; *Ponorac et al., 2020*), and 2.9–15% in male college or elite athletes (*Nabhan et al., 2020*; *Parks, Hetzel & Brooks, 2017*). Some researchers have attempted to provide iron supplements in athletes without anemia to improve their athletic performance. An Australian team reported that intravenous iron supplementation within 6 weeks of training improved the perceived fatigue and mood in distance runners without clinical ID (*Woods et al., 2014*).

In addition, the following mechanisms have been demonstrated to cause anemia in athletes: inadequate iron intake, intravascular hemolysis due to excessive exercise, gastrointestinal bleeding, exercise-induced acute phase response with production of inflammatory cytokines, dilution due to increased plasma volume, and insufficient intake of vitamins (*Mougios, 2007*; *Shaskey & Green, 2000*; *Williams, 1989*). These physiological responses have been suggested mainly in case reports or small retrospective studies, and the focus of current research was to verify their actual contribution to the development of anemia.

Most studies on anemia in athletes have focused on *professional* athletes or members of college athletic teams; however, little is known about the characteristics of anemia in young people who enjoy sports as part of school or community club activities. We have established an outpatient clinic specializing in anemia (*Takemoto, 2017*; *Takemoto, 2019*) to comprehensively assess the clinical laboratory data on anemia, including serum levels of haptoglobin, creatine kinase, folate, and vitamin B12 in addition to hemoglobin (Hb) and ferritin as a fundamental assessment for the athletes. This study aimed to clarify the clinical characteristics of anemia in adolescent athletes. We retrospectively reviewed the medical and laboratory records of middle school-aged to college athletes between 13 and 22 years old.

## MATERIALS AND METHODS

### Study participants and design

This study included patients between 13 and 22 years old who were affiliated with athletic clubs in schools and visited our outpatient clinic that specializes in the treatment of patients with anemia (Yoshinobu Clinic, Kagoshima, Japan) between August 1, 2016, and August 31, 2020. The participants visited our clinic due to complaints of anemia symptoms or following the advice from the trainers of their affiliated clubs. All the outpatients responded to the screening survey, including questions on sports club activities, at the first visit, which facilitated the selection of study participants.

The laboratory parameters measured using the blood samples of outpatients included the complete blood cell count and serum levels of ferritin, vitamin B12, folate, creatine phosphokinase, and haptoglobin. The serum zinc concentrations were measured after October 2019. Peripheral blood was collected and transported to a contract laboratory for measurement (Kagoshima City Medical Association Laboratory Center, Kagoshima, Japan). The patients answered the medical questionnaires containing questions related to their affiliated club, frequency of practice, menstruation (for female patients), and iron supplementation.

We retrospectively collected and analyzed the clinical data after approval from the Institutional Review Board (IRB) of the Medical Governance Research Institute (Tokyo, Japan) (approval number: MG2020-08-20200904). The requirement for obtaining informed consent was waived, and the consent was secured in an opt-out manner because of the retrospective and observational nature of this study in accordance with the Ethical Guidelines for Medical and Health Research Involving Human Subjects in Japan.
### Definitions

In our study, anemia was defined as an Hb level of ≤12.0 g/dL in both male and female patients aged ≤ 14 years, or lower than 13.0 g/dL in men aged > 15 years, in accordance with the criteria of the World Health Organization (WHO) (*WHO, 2011*). ID was defined as a serum ferritin level of ≤30 µg/L (*Camaschella, 2015*). The cutoff values of serum haptoglobin levels were 83, 66, and 25 mg/dL for phenotypes 1-1, 2-1, and 2-2, respectively, in accordance with the reference range of the testing laboratory. The cutoff values for serum levels of creatinine kinase, vitamin B12, and folate were 147 IU/L, 350 pg/mL, and 4 ng/mL, respectively. The mean corpuscular volume (MCV) and mean corpuscular hemoglobin (MCH) were calculated based on the complete blood cell count (*Reddy & Morlote, 2021*).

### Statistical analysis

The participants' characteristics were summarized using descriptive statistics. The two-sided 95% confidence interval (CI) of a proportion was determined using the Clopper–Pearson method. Two-group comparisons were performed using the Mann–Whitney U test for continuous variables and Fisher's exact test for categorical variables. Multivariate logistic regression models were employed to identify the patient characteristics associated with the risk of anemia. Multivariate analysis was performed using a stepwise backward selection with the Wald method for significant variables in the univariate assessment ($p < 0.05$). Statistical significance was considered when the two-sided *p value* was less than 0.05. All statistical analyses were performed using SPSS version 27 (IBM, Armonk, NY, USA).

## RESULTS

### Patient characteristics

The patient characteristics ($n = 485$ in total, 231 male (48%) and 254 female (52%) patients) are shown in Table 1. The median age was 15 years ([range,13–22] years). Nine females (4%) had not yet experienced their first menstruation, and 72 (28%) females reported irregular menstruation in the self-reported screening survey. The most common club membership was Track and Field/Athletics ($n = 171$, 35%), in which 101 (55%) practiced middle- or long-distance running. A total of 55 patients (11% of the total cohort) took either iron medication prescribed by physicians from other clinics or commercially available iron supplements (Fig. 1). We excluded them from further analysis and focused on non-treated individuals.

### Non-treated cohort

Anemia, as defined by the WHO criteria, was observed in 71 patients (16.5% (95% CI [13.1–20.4])), consisting of 18 males (9.0% (95% CI [5.4–13.8])) and 53 females (23.1% (95% CI [17.8–29.2])), in the non-treated cohort at initial examination ($n = 430$) (Table 2). No significant differences were observed in the patient characteristics between the anemic and non-anemic groups in both males and females. The anemia group, however, showed significantly higher proportions of MCV (80 fL), MCH (27 pg), and

Table 1 Participant characteristics.

| Variables | Median [range] or Number (percentage) |
|---|---|
| Age (years) | 15 [13–22] |
| Sex–female | 254 (52) |
| Height (cm)* | 163 (139–186) |
| Body weight (kg)* | 51 (34–92) |
| Body mass index (kg/m$^2$)* | 19.2 (13.6–31.1) |
| Affiliated club | |
| Track & Field/Athletics | 171 (35) |
| Basketball | 105 (22) |
| Soccer/football | 42 (9) |
| Volleyball | 27 (6) |
| Tennis | 26 (5) |
| Others | 114 (24) |
| Frequency of practice[†] | |
| Less than 5 times a week | 42 (8) |
| 5 times a week or more | 232 (48) |

**Note:**
Data were missing in 19[*] and 231† cases, respectively.

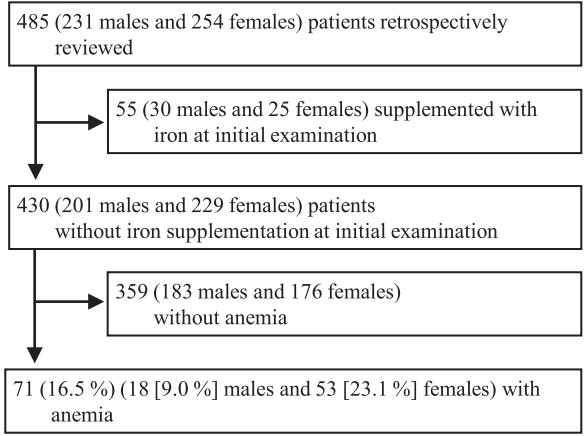

**Figure 1 Study cohort profile.** The procedures on inclusion of this study cohort is shown.

hypoferritinemia than the non-anemic group in both sex groups ($p = 0.001$, < 0.001, and < 0.001 for males, and < 0.001, < 0.001, and < 0.001 for females, respectively). Elevated serum creatinine kinase levels were more frequently observed in the anemic group than in the non-anemic groups in both men and women ($p = 0.002$ and 0.003, respectively). Anemic females exhibited a significantly higher prevalence of folate deficiency than non-anemic females ($p = 0.006$). No significant differences were observed in the prevalence of zinc deficiency between the anemic and non-anemic groups in either sex.

Iron deficiency was identified in 55 patients (77.5% in anemic patients) (13 male (72.2%) and 42 female (79.2%) patients). Then, the laboratory data of anemic patients with

**Table 2 Characteristics and laboratory data classified by sex and anemic status in non-treated participant group.**

| Variables | Male | | | Female | | |
|---|---|---|---|---|---|---|
| | Anemia (*n* = 18) | Non-anemia (*n* = 183) | *p* value | Anemia (*n* = 53) | Non-anemia (*n* = 176) | *p* value |
| *Patient Characteristics* | | | | | | |
| Age (years) | 16 [13–17] | 15 [13-22] | 0.300 | 15 [1–22] | 15 [13–22] | 0.414 |
| BMI (kg/m$^2$) | 18.6 [15.8–21.8] | 18.8 [15.0–31.1] | 0.449 | 20.3 [13.6–25.0] | 19.7 [13.7–29.8] | 0.933 |
| Menstruation | | | | | | 0.723 |
| No menstruation | | | | 1 (2) | 7 (5) | |
| Regular menstruation | | | | 31 (65) | 99 (65) | |
| Irregular menstruation | | | | 16 (33) | 47 (31) | |
| Affiliated club | | | 0.094 | | | 0.755 |
| Track & Field/Athletics | 11 (61) | 75 (41) | | 11 (21) | 36 (21) | |
| Basketball | 2 (11) | 17 (9) | | 21 (40) | 57 (32) | |
| Soccer/football | 4 (22) | 28 (15) | | 1 (2) | 3 (2) | |
| Others | 1 (6) | 63 (34) | | 20 (38) | 80 (46) | |
| Frequency of practice | | | 0.213 | | | 0.344 |
| Less than 5 times a week | 1 (14) | 20 (21) | | 3 (11) | 14 (17) | |
| 5 times a week or more | 6 (86) | 77 (79) | | 25 (89) | 71 (84) | |
| *Laboratory Data* | | | | | | |
| MCV (fL) | 87 [67–94] | 87 [75–96] | 0.199 | 84 [58–97] | 89 [75–97] | <0.001 |
| MCV <80 | 4 (22) | 2 (1) | 0.001 | 16 (30) | 2 (1) | <0.001 |
| MCH (pg) | 28 [20–32] | 30 [24–34] | 0.003 | 27 [15–32] | 30 [24–33] | <0.001 |
| MCH <27 | 7 (39) | 4 (2) | <0.001 | 22 (42) | 5 (3) | <0.001 |
| Reticulocyte (‰) | 11 [7–25] | 12 [5–21] | 0.369 | 13 [6–33] | 13 [6–30] | 0.715 |
| Reticulocyte >20 | 1 (6) | 1 (1) | 0.171 | 3 (6) | 9 (5) | 0.554 |
| Hypoferritinemia | 13 (72) | 35 (19) | <0.001 | 42 (79) | 66 (38) | <0.001 |
| Vitamin B12 deficiency | 9 (50) | 103 (56) | 0.394 | 25 (47) | 87 (49) | 0.448 |
| Folate deficiency | 0 (0) | 19 (10) | 0.139 | 10 (19) | 10 (6) | 0.006 |
| Elevation of serum creatine kinase level | 17 (94) | 110 (60) | 0.002 | 26 (49) | 48 (27) | 0.003 |
| Low haptoglobin | 6 (33) | 90 (49) | 0.150 | 21 (40) | 63 (36) | 0.363 |
| Zinc deficiency | 0 (0) | 19 (30) | 0.260 | 12 (71) | 30 (54) | 0.168 |

Notes:
Median [range] or number (percentage) are shown. No missing data were presented except for menstruation (201 females included), frequency of practice (104 males and 113 females) and Zinc deficiency (68 males and 73 females).
BMI, body mass index; MCV, mean corpuscular volume; MCH, mean corpuscular hemoglobin. The Mann-Whitney U tests and Fisher's exact tests were performed for continuous and for categorical variables, respectively.

and without ID were compared (Table S1). Significantly lower Hb, MCV, and MCH were found in the IDA group than in the non-IDA group in both sexes (*p* = 0.007, 0.03, and 0.003 for males, and 0.001, 0.003, and 0.001 for females, respectively). A significantly higher prevalence of vitamin B12 deficiency was observed in males with IDA than in males without IDA (69% *vs* 0%, *p* = 0.029). The Hb levels in anemic patients without ID in both sexes were close to the lower limits of the reference range.

**Table 3 Multivariate analysis to predict anemia.**

| Variables | Male | | Female | |
|---|---|---|---|---|
| | Odds | *p* value | Odds | *p* value |
| Hypoferritinemia | 13.2 [4.2–41.1] | <0.001 | 6.6 [1.3–13.9] | <0.001 |
| Elevation of serum creatine kinase level | 14.7 [1.8–118.4] | 0.012 | 2.7 [1.4–5.5] | 0.004 |

Note:
The adjusted odds ratios [95% confidence interval] are shown. The Nagelkerke R-squares of multivariate logistic models were 33.7% and 23.2% for male and female, respectively. The *p* values for the multivariate models were <0.001 for both sexes.

## Predictors for anemic status

Results of the univariate logistic regression analysis of the factors associated with the development of anemia in the male cohort revealed that belonging to the Track and Field/Athletics club, hypoferritinemia, and elevated creatinine kinase were significant predictors ($p = 0.036$, $p < 0.001$, and $p = 0.020$, respectively) (Table S2). Results of the multivariate analysis conducted in males revealed hypoferritinemia (odds ratio: 13.2 (95% CI [4.2–41.1]) for adjusted odds ratio, $p < 0.001$) and elevation of creatinine kinase level (odds ratio: 14.7 (95% CI [1.8–118.4]), $p = 0.012$) as independent predictors (Table 3). In females, hypoferritinemia and elevated creatinine kinase levels were also identified as independent predictors after conducting a multivariate regression analysis followed by a univariate assessment (6.6 [1.3–13.9], $p < 0.001$, and 2.7 [1.4–5.5], $p = 0.004$, respectively).

## Pre-treated patients with iron supplementation

A total of 55 participants, consisting of 30 males and 25 females, were treated with iron supplements, either using the prescribed medication from other outpatient clinics or over-the-counter (OTC) products before their initial presentation at our clinic. Twelve patients (six males and six females) exhibited symptoms of WHO-defined anemia despite receiving iron supplementation (Table 4). IDA was observed in seven of these patients (three males and four females). Eight patients (five males and three females) were administered with prescribed iron medication, while five presented with vitamin B12 deficiency.

## DISCUSSION

This study provides valuable information to physicians treating adolescent athletes since a set of comprehensive laboratory examinations including vitamin B12, Zn, haptoglobin, folate, and creatine kinase tests in addition to Hb and ferritin tests were performed as routine examinations. Notably, anemia was identified in 9% and 23% of males and females in our cohort, respectively. The prevalence of anemia in this study was higher than that in a previous study, in which 3.5% of boys and 8.1% of girls aged 14 to 17 years belonging in sports clubs were diagnosed with anemia (*Toivo et al., 2020*). Females are more susceptible to anemia than males due to menstrual blood loss. A few available studies have evaluated the prevalence of anemia in male athletes of middle and high school age. Anemia is an important issue in adolescent athletes, regardless of sex.

Table 4 Case presentation of individuals who were already taking prescribed iron medications or self-treating iron supplementation.

| Case | 1 | 2 | 3 | 4 | 5 | 6 | 7 | 8 | 9 | 10 | 11 | 12 |
|---|---|---|---|---|---|---|---|---|---|---|---|---|
| Sex | M | M | M | M | M | M | F | F | F | F | F | F |
| Type of pre-existing iron treatment | P | P | P | P | P | S | P | P | P | S | S | S |
| Age (years) | 14 | 15 | 16 | 16 | 16 | 15 | 14 | 14 | 16 | 13 | 14 | 17 |
| Hemoglobin (g/dL) | 11.5 | 12.2 | 12.8 | 12.8 | 12.5 | 12.8 | 11.6 | 11.5 | 9.8 | 8.4 | 9.8 | 10 |
| Ferritin (µg/L) | 7.7↓ | 68 | 50.1 | 22.3↓ | 22.4↓ | 83.1 | 110.1 | 9.9↓ | 67 | 3.2↓ | 6.7↓ | 7.2↓ |
| MCV (fL) | 80 | 91 | 91 | 89 | 85 | 94 | 92 | 87 | 86 | 73 | 81 | 90 |
| Vitamin B12 (pg/mL) | 230↓ | 458 | 291↓ | 229↓ | 215↓ | 254↓ | 225↓ | 519 | 336 | 840 | 395 | 695 |
| Folate (ng/mL) | 3.8↓ | 8.5 | 21.1 | 9.2 | 16.1 | 4.1 | 6.8 | 6.8 | 13 | 19 | 8.3 | 16.4 |
| Creatine Kinase (IU/L) | 187↑ | 297↑ | 450↑ | 222↑ | 157↑ | 2622↑ | 143 | 197↑ | 235↑ | 100 | 365↑ | 467↑ |
| Haptoglobin (mg/dL) | ≤10↓ | ≤10↓ | 42↓ | 68 | 15↓ | ≤10↓ | 100 | 26↓ | 30 | 29 | 15↓ | 21↓ |
| Haptoglobin-Type | N | N | 2-1 | 2-2 | 2-1 | N | 1-1 | 2-1 | 2-2 | 2-2 | 2-2 | 2-2 |
| Club membership | MMA | TFA | SF | TFA | TFA | SF | TFA | B | B | V | B | TFA |

Notes:
M, male; F, female; P, prescribed iron medication; S, self-treating over-the-counter iron supplementation; MCV, mean corpuscular volume; N, not identified for haptoglobin type; MMA, Mixed Martial Arts; TFA, Track & Field/Athletcs; SF, soccer/football; B, basketball; V, volleyball.
↓ lower than reference ranges for levels of serum ferritin, vitamin B12, folate and haptoglobin.
↑ higher than reference range for creatinine kinase.

ID is the most common cause of anemia in both males and females. In fact, 72% and 79% of anemic males and females, respectively, had IDA in this study. Insufficient iron intake, dietary restriction, menstruation, bleeding from the gastrointestinal tract, and iron loss through sweating are known causes of IDA in men and women, especially committed athletes (McClung, 2012). However, iron intake and iron loss through sweating or gastrointestinal bleeding were not evaluated in this study, which will be the focus of a future study to identify the determinants of IDA in adolescent athletes. Anemic participants *without* ID presented mild anemia close to the lower limit of our reference range: Hb levels of 12.8 g/dL (range: 11.7–12.9) and 11.8 g/dL (11.0–11.9) in males and females, respectively. This finding suggests that the clinical significance of anemia *without* ID may be minor; however, a hematopoietic disorder, such as aplastic anemia, should be ruled out.

Overtraining may be a risk factor for anemia in adolescent athletes. Approximately 40% of the participants presented low haptoglobin levels regardless of anemia and sex classification. Declining haptoglobin levels, in general, suggest the presence of chronic hemolysis, although serum free Hb levels as direct evidence of hemolysis were not assessed. In the anemia group, 94% and 49% of males and females, respectively showed elevated creatine kinase levels, while 86% and 89% of males and females, respectively practiced in the club for more than 5 days per week. The cause of hemolysis is exercise-induced intravascular hemolysis, which is commonly observed in long-distance runners (Lippi & Sanchis-Gomar, 2019). Excessive exercise is associated with bleeding in the digestive tract. The most common club membership in our cohort was Track & Field/Athletics, accounting for 35%. Among the Track & Field/Athletics club members, 59% specialized in middle- or long-distance running. These results support the association between chronic hemolysis and excessive training although the intensity and volume of their training

were lack to analyze in this study. Instructors of club sports should pay attention to the amount of practice to prevent overtraining, which may increase the risk of injury and anemia in young athletes.

Another possible contributor to the development of anemia in adolescent athletes is hepcidin response, which regulates iron homeostasis (*Camaschella, 2015*). Hepcidin stimulation decreases iron absorption in the intestine and promotes iron sequestration in tissues. An exercise-induced inflammatory response, which is indicated by elevated serum interleukin-6 levels, was associated with hepcidin expression (*Newlin et al., 2012*). Thus, alterations in hepcidin anemic athletes could be observed if hepcidin levels were measured. Future studies will include the assessment of hepcidin levels.

The sub-analysis of anemic patients showed the clinical importance of vitamin B12 metabolism when physicians consider the treatment of IDA. Approximately half of the participants, ranging between 47% and 56%, showed vitamin B12 deficiency across sexes and anemia conditions. Furthermore, a significantly higher proportion of males with IDA developed vitamin B12 deficiency compared with those without ID (69% *vs* 0%, $p = 0.03$). Vitamin B12 is an essential nutrient for the formation and maturation of red blood cells (*Oh & Brown, 2003*). Hemopoiesis due to exercise increases the demand for vitamin B12 (*Koury & Ponka, 2004*). Insufficient intake and dietary restrictions in athletes who are trying to control their weight may also cause vitamin B12 deficiency (*Economos, Bortz & Nelson, 1993*). Young athletes are at high risk of developing anemia due to vitamin B12 deficiency, in addition to ID.

Iron supplementation alone may not be sufficient for treating anemia in adolescent athletes. Interestingly, we identified vitamin B12 deficiency in five out of eight study participants who were already taking prescribed iron medications. Most OTC conventional iron preparations contain vitamin B12 and folic acid. Physicians commonly prescribe iron agents alone for the treatment of IDA. This study highlights the potential beneficial impact of vitamin B12 supplementation in addition to iron intake.

Our study can be considered significant as it reports the prevalence of anemia in young athletes by providing comprehensive clinical laboratory data; however, the study has some limitations. First, it was a retrospective analysis of the data on individuals who were considered or suspected of having anemia. Therefore, the results of this study cannot be generalized because they could overestimate the prevalence of anemia. Second, there may be unrecognized bias due to the small sample size. For example, many participants were affiliated with the Track & Field/Athletics club, soccer/football club, and basketball club. These athletes and their coaches may be paying more attention to anemia in light of recent news reports on the overtreatment of anemia with iron injection (*Kobayashi et al., 2019*).

## CONCLUSION

Anemia is common in both male and female young athletes. ID and the negative effects of intensive or excessive training, as indicated by elevated levels of serum creatinine kinase, could contribute to the development of anemia. Vitamin B12 deficiency may be a clinical pitfall for treatment. Physicians treating adolescent athletes who present with

anemia should pay attention to the amount and type of training and vitamin B12 metabolism, in addition to iron supplementation.

## ACKNOWLEDGEMENTS

We express our gratitude to medical and administrative staff at the Yoshinobu Clinic (Kagoshima, Japan) and Ms. Erika Yamashita (Medical Governance Research Institute, Tokyo, Japan) for their assistance of data collection and Editage service for the proofreading.

### Funding

This work was supported by Yoshinobu Clinic, Kagoshima, Japan. The funders had no role in study design, data collection and analysis, decision to publish, or preparation of the manuscript.

### Grant Disclosures

The following grant information was disclosed by the authors:
Yoshinobu Clinic, Kagoshima, Japan.

### Competing Interests

Kana Yamamoto is employed in part by Yoshinobu Clinic. Morihito Takita is employed by Navitas Clinic Tachikawa. Masahiro Kami is the director of the Medical Governance Research Institute and received a donation from Ain Holdings Inc. and remuneration for outside director of SBI Biotech Co., Ltd. Tetsuya Tanimoto received personal fees from MNES Inc. and Bionics Co. Ltd., outside of the submitted work and is employed by Navitas Clinic Kawasaki. Yoshinobu Takemoto is the director of Yoshinobu Clinic. All the other authors declare that they have no competing interests.

### Author Contributions

- Kana Yamamoto conceived and designed the experiments, performed the experiments, analyzed the data, prepared figures and/or tables, and approved the final draft.
- Morihito Takita analyzed the data, prepared figures and/or tables, and approved the final draft.
- Masahiro Kami conceived and designed the experiments, authored or reviewed drafts of the paper, and approved the final draft.
- Masaharu Tsubokura analyzed the data, authored or reviewed drafts of the paper, and approved the final draft.
- Tetsuya Tanimoto analyzed the data, authored or reviewed drafts of the paper, and approved the final draft.
- Toshio Kitamura analyzed the data, authored or reviewed drafts of the paper, and approved the final draft.
- Yoshinobu Takemoto conceived and designed the experiments, performed the experiments, authored or reviewed drafts of the paper, and approved the final draft.

## Human Ethics

The following information was supplied relating to ethical approvals (*i.e.*, approving body and any reference numbers):

We retrospectively collected and analysed the clinical data after approval of the Institutional Review Board of the Medical Governance Research Institute (Tokyo, Japan) (Approval number: MG2020-08-20200904).

## Data Availability

The raw data are available in the Supplemental File.

## Supplemental Information

Supplemental information for this article can be found online at http://dx.doi.org/10.7717/peerj.13004#supplemental-information.

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
