# Peer review of "Profiles of anemia in adolescent students with sports club membership in an outpatient clinic setting: a retrospective study"

_PeerJ, doi:10.7717/peerj.13004_

## Round 0.1 · original submission · Minor Revisions

Please make corrections according to the reviewers' suggestion or write a detailed rebuttal point by point.

Reviewer 1 ·

Basic reporting

It seems that the files that are uploaded are in the Edit format, meaning that the version that was submitted had all changes on so it was a bit hard to read the tables. Nevertheless I managed to turn that off even though I do not think that it is Ok to submit this kind of files as final version.

Experimental design

The design is clear and easy to follow. What we lack is the frequency and intensity and volume of their weekly engagement in sport

Validity of the findings

Basically i think that it is a bit of a novel approach and that many relevant factors were included as predictors. i think that the article lacks a physiology parts in discussion. Discussion mainly fokus on repeating the findings with no discussion about the physiology mechanisms of for example B12 and excercise etc...

·

Basic reporting

In general, the English writing is easy to read but there is still some errors in the text. For instance, on line 77, "...were reported as was 2-29% and 3-7%....." in which "as was" might be a typo. The recheck the English grammar throughout the paper.

The paper structure, content and references are finely written.

The hypothesis of the paper is well defined.

Experimental design

The experimental design is clearly described.
The methods is scientific and logical.

Validity of the findings

The study findings are relevant to the current knowledge and well answered to the study questions.

Additional comments

None

---

## Round 0.2 · accepted · Accept

The manuscript is after revision acceptable for publication.